# Peripheral Blood and Nasopharyngeal Swab MiRNA-155 Expression in Infants with Respiratory Syncytial Virus Infection

**DOI:** 10.3390/v15081668

**Published:** 2023-07-31

**Authors:** Francesco Savino, Stefano Gambarino, Maddalena Dini, Andrea Savino, Anna Clemente, Cristina Calvi, Ilaria Galliano, Massimiliano Bergallo

**Affiliations:** 1Early Infancy Special Care Unit, Regina Margherita Children Hospital, A.O.U. Città della Salute e della Scienza di Torino, 10126 Turin, Italy; francesco.savino@unito.it; 2Department of Public Health and Pediatric Sciences, Paediatric Laboratory, University of Turin, Medical School, 10136 Turin, Italy; gambarino.stefano@gmail.com (S.G.); maddalena.dini@edu.unito.it (M.D.); anna.clemente424@edu.unito.it (A.C.); cristina.calvi@unito.it (C.C.); ilaria.galliano@unito.it (I.G.); 3Post Graduate School of Pediatrics, Univeristy of Turin, 10124 Turin, Italy; andrea.savino@unito.it; 4Department of Pediatrics, Infectious Diseases Unit, Regina Margherita Children’s Hospital, University of Turin, Piazza Polonia 94, 10126 Turin, Italy

**Keywords:** miR-155, RSV, miRNA isolation, real-time PCR, bronchiolitis, swab, antiviral immunity

## Abstract

Introduction. MicroRNA (miR) 155 has been implicated in the regulation of innate and adaptive immunity as well as antiviral responses, but its role during respiratory syncytial virus (RSV) infections is not known. The objective of this study was to investigate the expression of miR-155 using pharyngeal swabs and peripheral blood in infants with RSV infection and uninfected controls. Methods. A prospective age-matched study was conducted in primary care in Torino from 1 August 2018 to 31 January 2020. We enrolled 66 subjects, 29 of them patients with RSV infection and 37 age-matched uninfected controls, and collected pharyngeal swabs and peripheral blood in order to assess miR-155 expression with real-time stem–loop–TaqMan real-time PCR. Results. The data show that there is no correlation between pharyngeal swabs and peripheral blood with respect to miR-155 expression. The 1/ΔCq miR-155 expression levels in throat swabs in RSV bronchiolitis patients and healthy controls were 0.19 ± 0.11 and 0.21 ± 0.09, respectively, and were not significantly different between healthy controls and bronchiolitis (*p* = 0.8414). In the peripheral blood, miR-155 levels were higher than those of healthy control subjects: 0.1 ± 0.013 and 0.09 ± 0.0007, respectively; *p* = 0.0002. Discussion. Our data provide evidence that miR-155 expression is higher in peripheral blood during RSV infection but not in swabs. This difference in the timing of sample recruitment could explain the differences obtained in the results; miR-155 activation is probably only assessable in the very early stages of infection in the swab and remains visible for longer in the blood. New investigations are needed in order to clarify whether the miR-155 expression in swabs can be influenced by different stages of virus disease of infants.

## 1. Introduction

Respiratory syncytial virus (RSV) is the main cause of respiratory infections and of hospitalization for bronchiolitis among infants younger than 12 months [1]. Worldwide, RSV causes nearly 34 million lower respiratory tract infections (LRTI) and 3.4 million hospitalizations per year in infants and children younger than 5 years, with an estimated annual increase of 10% [2,3]. RSV is currently a public health problem in many countries, including Italy [4]. RSV was previously included in the subfamily Paramixoviridae but was recently reclassified in the family Pneumoviridae [5]; the virus is characterized by a large envelope and negative-sense RNA encoding 11 glycoproteins. RSV infections are seasonal, peaking during the winter months in temperate regions [6,7]. Young children, the elderly, and people with chronic diseases are at greatest risk for severe RSV infections [8,9].

MicroRNAs (miRNAs) are small non-coding RNAs that inhibit translation of mRNA to protein by binding to a specific target mRNA [10]. Several studies have investigated miRNA responses to RSV in vitro [11,12]. miR-155 expression has been shown to be elevated following influenza A and SARS-CoV-2 viral infection [13,14]. In addition, miR-155 has been shown to be remarkably upregulated during vesicular stomatitis viral infection, and it promoted type I interferon signaling [15], which is potentially pathogenic in influenza infection, according to a previous study [16]. Viruses can induce the up-/downregulation of certain host miRNAs to evade the host’s immune system, enabling the defense mechanisms to combat the infection [17,18].

Previous data on mechanisms regulated by miRNAs suggest a likely role in RSV infection; miRNAs are involved in the regulation of immune and/or inflammatory pathways at different levels [19,20,21], and for this reason, studies in this area are needed to better elucidate the interaction between RSV and host, especially in the first month of life. In this context, gene regulation mechanisms could represent an innovative way to identify potential biomarkers of infection in order to develop antiviral therapeutics for certain diseases for which no real treatment is yet available, such as RSV infection. Studies in human infants have found that during naturally occurring rhinovirus (RV) or respiratory syncytial virus (RSV) infections, the airways produce abundant miR-155. However, the role of miR-155 in viral respiratory infections remains an important unresolved paradox. Although miR-155 is essential for the generation of host TH1 antiviral immunity, it may also contribute to respiratory disease by enhancing allergic TH2 responses and NFkB-mediated inflammation in macrophages and other bone marrow-derived immune cells [21].

Over the past decade, noncoding RNAs (ncRNAs) and miRNAs have emerged as novel tools for medical decision making. They are easily measured by standard techniques already used in clinical laboratories, such as RT-qPCR. miRNAs are sensitive, robust, and inexpensive biomarkers that provide additional information to already established clinical variables and clinical indicators [22]. They are stable in various body fluids and offer advantages as biomarkers because they are highly conserved among different species and their expression patterns are tissue- and life-phase-specific.

To achieve the aims of the present work, we chose miR-155 with previously validated involvement in infected processes to examine its expression in peripheral blood and pharyngeal swabs from patients with RSV virus infection.

## 2. Methods

### 2.1. Subjects

This prospective case-control study was conducted in Turin, Italy, between 1 August 2018 and 31 January 2020. We included healthy, full-term infants hospitalized at Regina Margherita Children Hospital, Turin, Italy, for their first episode of bronchiolitis. Controls were uninfected healthy infants younger than 12 months who attended an outpatient clinic in the Department of Pediatrics for routine postpartum examinations.

Bronchiolitis was diagnosed by trained pediatricians based on clinical signs such as wheezing with or without cough, rales, dyspnea, increased respiratory rate, and retractions of the respiratory muscles. Infants hospitalized with bronchiolitis underwent routine blood tests during their recovery.

The study and the data collection procedure were approved by the Ethics and Research Committee of the Città della Salute e della Scienza di Torino on 24 November 2014, with prot. number 116918. Informed consent was obtained verbally from the parents of the study participants and consigned to their clinical records in accordance with the Italian good clinical practices and hospital clinical investigations guidelines. The samples were anonymized before processing.

The mean age of the 29 infants (45% male and 55% female) infected and examined with RSV was 85 (9–146) days at hospital admission. Their mean gestational age at birth was 38 weeks and their mean birth weight was 3100 g (2780–3730). The 34 infants in the control group (51.6% boys) were admitted with a mean age of 93 (4–48) days, and 41.9% were still exclusively or predominantly breastfed. Their mean gestational age at birth was 37 weeks, and their mean birth weight was 3020 g (2560–3980). They had not been hospitalized for bronchiolitis or other infections. White blood cells, neutrophil granulocytes, and eosinophil granulocytes were obtained from the medical records.

We included age-matched control subjects without viral respiratory infection (with negative viral PCR) recruited during nonrespiratory hospitalizations or outpatient clinic/emergency department visits (*n* = 37). The characteristics of the study participants are shown in Table 1.

RSV was diagnosed using sensitivities and rapid Antigen Xpert Xpress FLU/RSV (Cepheid, Sunnyvale, CA, USA) at Laboratory of A.U.O. Città della Salute e della Scienza di Torino without molecular confirmation.

The exclusion criteria for the patients and controls included known or suspected premature birth at less than 37 weeks of gestation, confirmed or suspected immune defects (e.g., primary immunodeficiencies, hematological diseases/neoplasms born to HIV+ mothers or other immunodeficiency-inducing diseases), cancers, viral coinfections, autoimmune disorders, food allergy, prematurity, autism spectrum disorder.

### 2.2. miR Isolation, cDNA Synthesis, and Real-Time PCR

For each throat swab (1 mL) and heparinized blood (0.2 mL) sample, RNA was extracted according to the simply RNA Blood Kit protocol without modification in the Maxwell16 system (Promega, Madison, WI, USA) according to the manufacturer’s instructions. RNA was eluted in a final volume of 50 μL and stored at −80 °C until use. Reverse transcription (starting from 500 ng of total RNA) was performed using the Gene Amp RNA PCR kit (Life technologies, Carlsbad, CA, USA) with modifications: 50 U MMLV RT, 1× PCR buffer II, 1 mM dNTPs, 5 mM MgCl2, 1 U RNase inhibitor, and 0.5 µg miR-155 stem loop primer (SLP) (Table 2) as previously described by the authors [23,24,25]. The reaction was performed in three steps: a first incubation at 16 °C for 30 min, another at 42 °C for 1 h, and a final incubation at 99 °C for 5 min. Purity and concentration of RNA were determined spectrophotometrically using NanoDrop ND-2000 (Thermo Fisher Scientific, Wilmington, DE, USA). After reverse transcription, asymmetric PCR was done using 300 nM specific forward primer, 4 μL 5× Colorless GoTaq^®^ Flexi Buffer, 0.1 U GoTaq^®^ Hot Start Polymerase (Promega, Bergamo, Italy), and 2 μL cDNA, resulting in a final volume of 20 μL. The thermal profile was as follows: 95 °C for 2 min; 30 cycles of 94 °C for 15 s, 55 °C for 30 s, and 72 °C for 20 s 5 μL of the enriched cDNA, referred to as ccDNA, was added to 35 μL of reaction mix containing 200 nM MGB probe, 1000 nM universal reverse primer, 800 nM forward primer (Table 2), and 1× GoTaq PCR Master Mix (Promega) in a final volume of 40 μL. Amplifications were executed on the ABI 7500 Real-Time PCR System (Life Technologies, Carlsbad, CA, USA) in a 96-well plate at 95 °C for 10 min followed by 40 cycles of 95 °C for 15 s and 60 °C for 1 min. Each sample was run in triplicate. The data were acquired in auto Ct mode and ΔCq was calculated by subtracting the Cq value of RNU43 RNA from the Cq value of miRNA of interest [23,24,25].

### 2.3. Statistical Analysis

The Spearman correlation test was performed to estimate the correlations concerning transcription levels of mir-155 in every sample analyzed (pharyngeal swab and peripheral blood). The Mann–Whitney test was used to compare the mir-155 expression in pharyngeal swabs and peripheral blood of children with bronchiolitis and those of the healthy control population. Statistical investigations were performed using the Prism software (GraphPad Software, La Jolla, CA, USA). In all analyses, *p* < 0.05 was considered statistically significant.

## 3. Results

All the patients were previously screened for the RSV infection, based on the medical guidelines for patient healthcare.

miR-155 was detected in all 66 samples (29 RSV-positive infants and 37 healthy controls) of throat swabs and peripheral blood.

Based on relative quantification of 1/ΔCq data, we correlated miR-155 expression in pharyngeal swabs and peripheral blood of all 66 samples using a Spearman test (*p*-value = 0.5361) (Figure 1). These data showed that there was no correlation between pharyngeal swabs and peripheral blood with respect to miR-155 expression.

Throat swab and peripheral blood miR-155 expression was compared in RSV-positive bronchiolitis patients and healthy uninfected controls.

The 1/ΔCq miR-155 expression levels in throat swabs of RSV bronchiolitis patients and healthy controls were (median age and 25–75% IQR): 0.177, 0.14–0.25, and 0.175, 0.13–0.24, respectively. miR-155 expression levels in RSV bronchiolitis patients were not significantly different between healthy controls and bronchiolitis (*p* = 0.8414), as shown in Figure 2.

The 1/ΔCq levels of miR-155 expression in the peripheral blood of RSV bronchiolitis patients were higher than those of healthy control subjects (median age and 25–75% IQR): 0.1, 0.09–0.12 and 0.09, 0.09–0.1, respectively; *p* = 0.0002 (Figure 2).

## 4. Discussion

miRNAs are paving the way for their routine use in the clinical diagnosis and prognosis of a variety of diseases, such as viral respiratory infections. However, the increasing acceptance and use of molecular-based miRNA assays for research and diagnostic purposes requires rigorous validation before they are used in clinical diagnosis. Several RT-qPCR assays have been developed for miRNA quantification and are currently being made available by companies. Considering the similar small size of miRNAs with common PCR primers, we proposed an optimal alternative method based on stem loop primer [23,24,25].

Studies have shown that miRNAs can be stably detected in saliva samples [26,27,28,29].

Numerous published studied have confirmed the important role of miR-155 in viral infections [13,14]. For example, overexpression of miR-155 led to significant reduction in human HIV replication in macrophages [30]. It has been described that miR-155 regulates viral infections caused by Epstein–Barr [31], Borna disease [32], and reticuloendotheliosis viruses [33,34]. miR-155 suppresses Japanese encephalitis virus (JEV) replication in microglial cells and regulates JEV-induced inflammatory response in mouse brains [35,36]. Studies in human infants [37,38] have identified abundant airway production of miR-155 during naturally occurring infections caused by rhinovirus (RV) or respiratory syncytial virus (RSV). miRNA-155 is well-studied in patients suffering from severe coronavirus disease 2019 (COVID-19) [39].

In agreement with Arroyo et al. [21], in our study we observed increased levels of miR-155 in the peripheral blood of RSV-infected infants compared with healthy controls, but found no significant difference in throat swabs. Arroyo and colleagues collected nasal secretions at the onset of acute respiratory illnesses, whereas RSV infections in our patients were present at least since the third day after hospital admission, which is why we examined the throat swabs at recovery and not on the first day or in the initial phase of viral infections. This difference in the timing of sample recruitment could explain the differences obtained in the results. miR-155 activation is probably only assessable in the very early stages of infection in the swab and remains visible for longer in the blood.

Arroyo et al. [21] reported recently that the airway secretion of miR-155 during viral respiratory infections in young children is associated with enhanced antiviral immunity (Th1-type polarization) in subjects younger than 2 years. Monitoring miRNA levels in pharyngeal swabs during the course of infection might be of great importance, because of their stability in biological fluids and frequent alterations during both chronic and acute airway infections [40]. Probably, the results found in the swab can be used to obtain prognostic value. It can be hypothesized that the expression kinetics of miRNAs is different in the different districts of the body. In the swab, this expression probably reflects a non-inflamed state, synonymous with overcoming the infection, a fact not yet visible in the peripheral blood. As evidence of this fact, none of our patients had a poor prognosis and all survived the infection without sequelae.

We know that upregulation of miR-155 is associated with an acute inflammatory response [41]. Dendritic cells express this miRNA as well as T and B lymphocytes, and is required for their function [42]. miR-155 is produced upon TLR4-mediated NF-κB activation and positively regulates myeloid proliferation and dendritic cell maturation and lymph nodes’ migration [43].

miR-155 is an important modulator of both innate and adaptive immune responses and plays a critical role in viral and parasitic infections, enhancing mammalian host defense mechanisms against pathogens [44,45]. Numerous miRNAs are involved the extensive and intricately coordinated processes and interactively affect multiple regulatory pathways [46]. miRNA was recently shown to contribute to the development of fatal acute respiratory distress syndrome (ARDS) in H1N1 influenza A virus-infected mice, suggesting that it may be a valuable biomarker for tracking individual patient immune responses [47].

To our knowledge, this is the first study of miR-155 expression using both throat swabs and peripheral blood in hospitalized infants with RSV infection. As a limitation of the study, further studies are needed to confirm our data and also new studies with subjects infected with RSV at different ages to clarify whether miR-155 expression may be related to the timing of viral infection. Finally, new research on different miRNAs in RSV disease may provide more information on the mechanisms underlying the link between host immunity and virus. However, our approach to miR-155 provides an understanding of the pervasive effects of this multifunctional miRNA in health and disease and highlights the need for profiling circulating miRNAs to identify clusters and signatures that will aid in patient stratification and treatment.

## 5. Conclusions

In summary, we found an upregulation of miR-155 in peripheral blood in infants with acute RSV infections at recovery in hospital compared to age-matched uninfected healthy controls, but not in pharyngeal swabs.

Because miR-155 signaling is a sensitive pathway of immunomodulation and can modulate the innate immune system, it is not possible at this time to confirm our hypothesis that expression of miR-155 in the smear is a negative prognostic factor because we did not have patients with a severe prognosis in our series. We plan to expand the panel of miRNAs to be studied and include a larger and more complete casuistry to investigate the prognostic role and possible therapeutic targets.

## Figures and Tables

**Figure 1 viruses-15-01668-f001:**
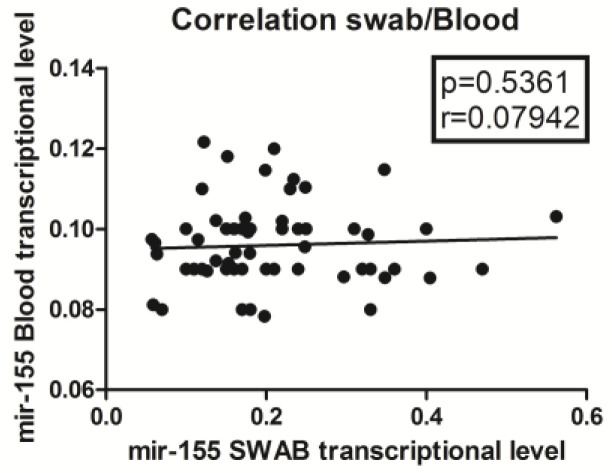
Correlations between transcription levels of mir155 in whole blood and nasopharyngeal swabs: relative quantification. Circles show the mean of three individual measurements. Line: linear regression line. Statistical analysis: Spearman correlation test.

**Figure 2 viruses-15-01668-f002:**
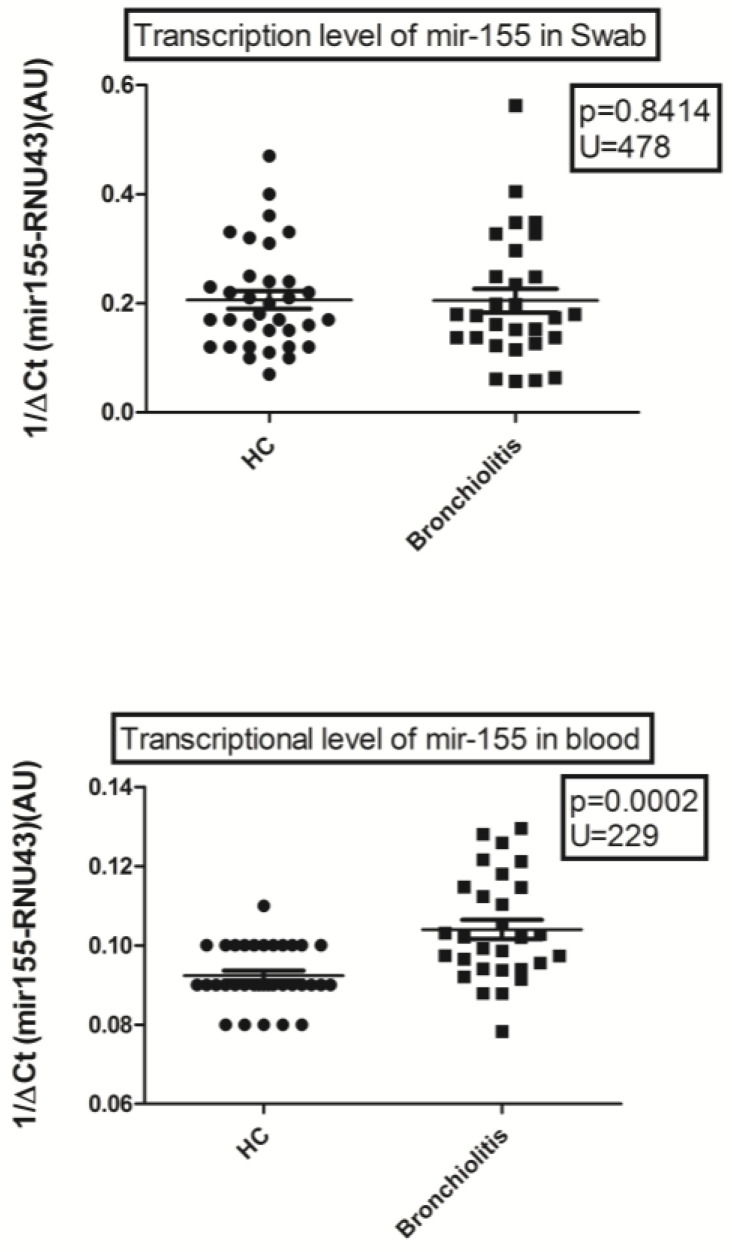
Expression of mir155 in nasopharyngeal swabs and whole blood from 29 patients with RSV-bronchiolitis and 37 healthy controls (HC). 1/DCt: relative quantification. Circles and squares show the median of three individual measurements, horizontal lines the median values. Statistical analysis: the Mann–Whitney test was used to compare the transcriptional levels of mir155 between children with bronchiolitis and control children.

**Table 1 viruses-15-01668-t001:** Characteristics of the two study groups of infants involved in the study: those infected by RSV and healthy controls.

	Infected by RSV (*n* = 29)	Healthy Controls (*n* = 37)	*p*-Value
Type of delivery:			
vaginal/caesarean	18/11	22/15	N.S.
Age at enrolment (mean days, range)	85 (9–146)	93 (9–48)	N.S. *
Gender			
Male, n (%)	13 (46)	19 (51)	N.S. ^#^
Female n (%)	16 (54)	18 (49)	N.S. ^#^
Birth weight (g, range.)	3500 (2780–3730)	3020 (2560–3980)	N.S. *
Gestational age (wks ± s.d.)	38 ± 2	37 ± 1.5	N.S. *

^#^ Fisher’s test, * Mann–Whitney test.

**Table 2 viruses-15-01668-t002:** Primers and probe list. All the sequences are reported in 5′–3′ direction.

**Target**	**RNU43**
sequence	GAACUUAUUGACGGGCGGACAGAAACUGUGUGCUGAUUGUCACGUUCUGAUU
SLP	GGCTCTGGTGCAGGGTCCGAGGTATTCGCACCAGAGCCAATCAG
Forward	TGACGGGCGGACAGAAA
Probe MGB fam	TGTGTGCTGATTGTCA
**Target**	**miR-155**
sequence	UUAAUGCUAAUCGUGAUAGGGGU
SLP	GGCTCTGGTGCAGGGTCCGAGGTATTCGCACCAGAGCCACCCCT
Forward	CGCAGTTAATGCTAATCGTGATA
Probe MGB fam	GGGGTGGCTCTGG
Universal reverse primer	TGCAGGGTCCGAGGTATTC

## Data Availability

Not applicable.

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
