# Peer review of "Peripheral Blood and Nasopharyngeal Swab MiRNA-155 Expression in Infants with Respiratory Syncytial Virus Infection"

_viruses, 2023, doi:10.3390/v15081668_

Round 1

Reviewer 1 Report

I congratulate the authors, the work is well structured, interesting and scientifically valid. I consider it very interesting given the considerable diffusion of RSV infections, especially in children, and the impact that RSV infections have on the healthcare system in terms of costs.

I have only a few minor considerations.

1.       The authors state (page 3) that the diagnosis of RSV infection was made by antigen testing. I would like to know if the diagnosis was confirmed or not with a subsequent RT-PCR molecular test, if so, I would suggest clarifying this aspect in the text.

2.       Also on page 3, the authors state that the impairment of immunological function was one of the exclusion criteria. I would suggest the authors to list the type eg: immunodeficiencies, haematological diseases/neoplasms or other immunodeficiency-inducing diseases.

3.       On page 3 in the materials and methods section I would suggest the authors add the volume (microlitres) of blood or medium of nasopharyngeal swab from which they started to extract the RNA.

4.       I would suggest to the authors to add in the graph of the swab/blood correlation of figure 1 in addition to the p value also the value of Spearman r value by adding it directly on the graph (even if there is no correlation).

5.       In the discussion section, the authors underline how MiR-155 is upregulated during the immune response and in particular following some viral infections and how important its function is in contrasting viral infections and more generally in innate and adaptive immunity. It would be important, if possible, if the authors could evaluate (perhaps on the blood samples of the subjects studied) the expression of genes associated with the immune/inflammatory response that appear regulated by MiR-155 (such as FADD or SOCS-1 or MYD88 or SPI1 or TNF) as it would be interesting and would enrich the work. Of course if that is feasible.

Author Response

Reviewer 1

I congratulate the authors, the work is well structured, interesting and scientifically valid. I consider it very interesting given the considerable diffusion of RSV infections, especially in children, and the impact that RSV infections have on the healthcare system in terms of costs.

I have only a few minor considerations.

  1. The authors state (page 3) that the diagnosis of RSV infection was made by antigen testing. I would like to know if the diagnosis was confirmed or not with a subsequent RT-PCR molecular test, if so, I would suggest clarifying this aspect in the text.

We clarify it in the text.

  1. Also on page 3, the authors state that the impairment of immunological function was one of the exclusion criteria. I would suggest the authors to list the type eg: immunodeficiencies, haematological diseases/neoplasms or other immunodeficiency-inducing diseases.

We correct the sentences.

  1. On page 3 in the materials and methods section I would suggest the authors add the volume (microlitres) of blood or medium of nasopharyngeal swab from which they started to extract the RNA.

We clarify it in the text.

  1. I would suggest to the authors to add in the graph of the swab/blood correlation of figure 1 in addition to the p value also the value of Spearman r value by adding it directly on the graph (even if there is no correlation).

We add data in the figure.

  1. In the discussion section, the authors underline how MiR-155 is upregulated during the immune response and in particular following some viral infections and how important its function is in contrasting viral infections and more generally in innate and adaptive immunity. It would be important, if possible, if the authors could evaluate (perhaps on the blood samples of the subjects studied) the expression of genes associated with the immune/inflammatory response that appear regulated by MiR-155 (such as FADD or SOCS-1 or MYD88 or SPI1 or TNF) as it would be interesting and would enrich the work. Of course if that is feasible.

Thank you for the idea. We also read the recent paper "Luteolin inhibits respiratory syncytial virus replication by regulating the MiR-155/SOCS1/STAT1 signaling pathway. Wang S, Ling Y, Yao Y, Zheng G, Chen W. Virol J. 2020 Nov 25;17( 1):187. doi: 10.1186/s12985-020-01451-6." in this regard and we believe that it can be an excellent starting point for future work. Given the scarcity of the sample we are unable to test these targets on the samples covered by the work.

Reviewer 2 Report

Estimated Authors,

I'm gratulating with you for this very interesting and well written paper on basic research about RSV and the up/down regulation of miRNA after RSV infection.

From my point of view, the present paper could be accepted for publication after some minor adjustments, and namely:

1. Statistical analysis:

a) please provide the rho value of all correlation analyses you did perform

b) similarly, please provide U value for your M-W test;

c) when reporting your data, as you did perform non-parametric analyses, please provide range and 95%CI rather than SD

d) please provide an appropriate unit of measure for the copies of miRNA (e.g. copies?)

e) please update your XY graphs with more accurate labels (e.g. Blood --> transcription levels of miRNA155 in Blood.

2. Could you provide CT (cycle thresholds) values for qPCR analyses?

Only minor typos scattered across the main text.

Author Response

Reviewer 2

I'm gratulating with you for this very interesting and well written paper on basic research about RSV and the up/down regulation of miRNA after RSV infection.

From my point of view, the present paper could be accepted for publication after some minor adjustments, and namely:

  1. Statistical analysis:
  2. a) please provide the rho value of all correlation analyses you did perform

We insert  it in figure 1.

  1. b) similarly, please provide U value for your M-W test;

This data is present in the figure 2.

  1. c) when reporting your data, as you did perform non-parametric analyses, please provide range and 95%CI rather than SD

we correct as requested.

  1. d) please provide an appropriate unit of measure for the copies of miRNA (e.g. copies?)

we use a relative quantification and therefore these are arbitrary units (AU).

  1. e) please update your XY graphs with more accurate labels (e.g. Blood --> transcription levels of miRNA155 in Blood.

We upgrade all graphs.

  1. Could you provide CT (cycle thresholds) values for qPCR analyses?

We insert the data in the text.

Reviewer 3 Report

Viruses-2528641

The present manuscript provides data on the expression of miR-155 in peripheral blood, and nasal swabs from children with or without RSV bronchiolitis. However, the work has several limitations in the experimental approach and it will benefit from an additional round of editing. Specific comments follows:

Major comments:

1. The major concern for this work is lack of novelty since the expression of miR-155 in infants infected with RSV has been reported in nasal secretions previously. Moreover, the authors do not discuss appropriately the discrepancy with their work.

2. The manuscript lacks RSV-specific discussion on the importance of miR-155 in RSV infection. The information included in the introduction is very scarce. The discussion in that respect is vague and does not justify the focus on RSV.

3. In the results section, the authors indicate that “All patients were previously screened for the RSV infection…”. However, the work is missing the information on how many of the tested samples were infected only with RSV and how many were infected with mixed co-viral respiratory infections.  That information must be shown.

4. Additional data are missing to understand the level of severity of the infection in the children tested and whether there is a correlation with the levels of the miRNA expression.

5. The work needs to show the analysis of other miRNAs to compare the expression to miR-155. Is miR-155 the predominant miRNA? How that compares to other miRNAs?

Minor comments:

6. The manuscript needs an additional round of editing to eliminate typos and to ensure all the manuscript is in English, e.g. line 2 in the abstract should be syncytial no syncyzial.

No major concerns on the English language. Although the manuscript will benefit from an additional round of editing.

Author Response

Reviewer 3

The present manuscript provides data on the expression of miR-155 in peripheral blood, and nasal swabs from children with or without RSV bronchiolitis. However, the work has several limitations in the experimental approach and it will benefit from an additional round of editing. Specific comments follows:

 Major comments:

  1. The major concern for this work is lack of novelty since the expression of miR-155 in infants infected with RSV has been reported in nasal secretions previously. Moreover, the authors do not discuss appropriately the discrepancy with their work.

We have increased the discussion and tried to answer your requests.

  1. The manuscript lacks RSV-specific discussion on the importance of miR-155 in RSV infection. The information included in the introduction is very scarce. The discussion in that respect is vague and does not justify the focus on RSV.

We have added a sentence in the introduction to justify the scope of the work and the link between miR-155 and RSV.

  1. In the results section, the authors indicate that “All patients were previously screened for the RSV infection…”. However, the work is missing the information on how many of the tested samples were infected only with RSV and how many were infected with mixed co-viral respiratory infections. That information must be shown.

We insert in material and methods this data. All the co-infection were excluded.

  1. Additional data are missing to understand the level of severity of the infection in the children tested and whether there is a correlation with the levels of the miRNA expression.

The definition of severity in the case of RSV infection is still highly debated today. Our series includes all subjects in similar physical conditions and no patients in intensive care.

  1. The work needs to show the analysis of other miRNAs to compare the expression to miR-155. Is miR-155 the predominant miRNA? How that compares to other miRNAs?

We thank you for this comment but at the moment we are unable to respond to this request. the mirna field is sufficiently complex and your suggestion can be accepted for future studies.

Minor comments:

  1. The manuscript needs an additional round of editing to eliminate typos and to ensure all the manuscript is in English, e.g. line 2 in the abstract should be syncytial no syncyzial.

We revised as suggested.